# Effects of a choral program combining wind instrument performance and breathing training on respiratory function, stress, and quality of life in adolescents: A randomized controlled trial

**Byeong Soo Kim[1], Ho Kim[1], Ji Youn Kim[2]***

1 Department of Physical Therapy, Graduate School, Daejeon University, Daejeon, South Korea,
2 Department of Fusion in Performing Art, College of Design Art, Daejeon University, Daejeon, South Korea

☯ These authors contributed equally to this work.
* jymusic@dju.kr

## Abstract

### Background

Choral activities are correlated with various health and wellbeing parameters. However, an intervention combining a music program using wind instruments and choral activities has not yet been investigated. Thus, this study aimed to assess the effects of a 12-week intervention combining a wind instrument performance program and a choral program on stress factors, quality of life, and respiratory function in adolescents located in a metropolitan city with exposure to air pollution.

### Method

This randomized controlled trial consisted of 50 adolescents, and the subjects were randomly assigned to a combination wind instrument and choral training group, a choral training group, and a control group. Following a 12-week intervention program, respiratory function, stress factors, and quality of life were compared between the three groups.

### Results

Regarding respiratory function, with the exception of maximal inspiratory pressure, all measured variables exhibited an interaction to indicate a variation in the pattern of change (p<0.05). Furthermore, regarding stress factors and quality of life, all measured variables exhibited an interaction to indicate a variation in the pattern of change(p<0.05). As a result of the post-hoc analysis, significant differences were found in all variables in experimental group 1 compared to other groups (p<0.05).

### Conclusion

The results showed that the 12-week intervention combining a wind instrument performance program and a choral program had positive effects in improving the respiratory function,

**Data Availability Statement:** Since the subjects of the research data are adolescents, the consent of

the institution entrusted with the consent of the guardian is required. The authority for this belongs to the Daejeon University Institutional Life Ethics Committee(+82-42-280-2029, dudtls4654@dju. kr). When requesting research data, if you send the exact reason why you need research data, only data excluding personal information (subject's date of birth, height, age, weight, and name) will be disclosed.

**Funding:** This research was supported by the National Research Foundation of Korea(NRF) grant funded by the Korea government(MSIT) (No.2019R1G1A110054313). The funders had no role in study design, data collection and analysis, decision to publish, or preparation of the manuscript.

**Competing interests:** The authors have declared that no competing interests exist.

stress factors, and quality of life in adolescents. This study findings are expected to support future studies aimed at promoting overall health including respiratory function and psychological factors through various music-based programs.

## Introduction

With the expansion of industrialization and population concentration in urban areas, the use of fossil fuels has increased, with the excessive release of various pollutants resulting in severe damage to the natural environment. Air pollution, a form of environmental pollution, involves the release of pollutants that exceed the self-purification capacity of the atmosphere; it can cause the loss of property loss and damage to numerous people, animals, and plants in specific regions. In general, mild damage due to air pollution can be the result of natural factors such as volcanic eruption, forest fires, animal and plant activities, and geographical conditions; however, the damage due to a spectrum of air pollutants from artificial factors due to daily activities and industrial activities by humans has led to a far more serious problem [1].

The impact of air pollution on health can be illustrated by the negative effect of long-term exposure to ambient particles, including nitrogen oxides ($NO_2$) and inorganic acid vapor, on the development of pulmonary functions [2, 3]. Increased air pollution may be related to reduced expiratory flow (forced expiratory volume in the first second; FEV1, forced vital capacity; FVC, maximal mid-expiratory flow; MMEF, and forced expiratory flow at 75% of FVC; FEF75), increased frequency of asthma aggravation, and reduced respiratory capacity in children [4]. A study on 249 high school students in New York found a significant positive correlation between the onset of acute respiratory symptoms and the concentration of substances related to particulate matter (PM) [5]. In a study using a time-stratified case crossover design, where the national registry data of Denmark and the air pollution data were combined (2001–2008), the cause of hospitalization of 8,226 children and adolescents aged 0–18 years was asthma [6]. Furthermore, in a longitudinal study on 2,444 pediatric cancer patients aged 0–14 years and adolescent and younger adult cancer patients aged 15–39 years with a diagnosis in 1986–2015, the increase in PM containing micro-particles of < 2.5 μm diameter was associated with the mortality of pediatric, adolescent, and younger adult patients with specific cancer types [7].

Recently, studies have investigated a diversity of physical therapy interventions, including gait exercise, core exercise, respiratory muscle strengthening, complex breathing training, and other such programs, to enhance respiratory function [8, 9]. To increase participant interest, interventions based on leisure activities have recently been introduced, and among such interventions, various programs based on music have been suggested. Choral activities are correlated with various health and wellbeing parameters including mood improvement, relaxation, and enhanced respiratory capacity [10]. Breathing training programs based on wind instruments have been shown to exert a positive effect on pulmonary function, cardiopulmonary endurance, and quality of life [11]. As an adjunct therapy for pediatric asthma, music-based treatments have been reported to greatly enhance the patients' pulmonary function and reduce hospitalization to assist with an improvement of quality of life [12, 13]. Therefore, intervention programs based on music and musical instruments have diversified with verified effects. However, no study has yet investigated an intervention combining a music program using wind instruments and that using choral activities.

This study thus aimed to determine the effects of a 12-week intervention combining a wind instrument performance program and a choral program on stress factors, quality of life, and

respiratory function in adolescents attending local schools in a community located in a metropolitan city with exposure to air pollution.

## Materials and methods

### Participants

By ANOVA: Repeated measures within-between interaction, effect size f = 0.25, the α (type I) error level = 0.05, the statistical power (or the β (type II) error level) = 0.80, number of groups = 3, With Number of measurements = 2, correlations among pairs of the repeated measurements = 0.5, and nonsphericity correction ε = 1 were set, and the total sample size was 42 people, Considering the dropout rate of 20%, 50 participants were recruited. The study participants included 50 local middle school and high school students in a community in D metropolitan city in South Korea. The inclusion criteria were adolescents aged 14–17 years, high-risk stress with psychological social well-being index short form (PWI-SF) scores ≥ 27, without a chronic respiratory disease, no hospitalization history due to an acute respiratory disease within the past 2 months, no COVID-19 history, and agreed to participate in this study with consents from their guardians. The exclusion criteria were individuals with cardio-cerebrovascular disease, auditory or vocal structural abnormalities that prevent fluent communication, history of surgery related to the respiratory system, and significantly reduced cognitive function due to disability or mental disorder [13]. During the screening process, all subjects satisfied the inclusion and exclusion criteria. A total of 50 subjects participated in the study. This study was conducted with the approval of the Institutional Review Board at Daejeon University. After obtaining written consent from the guardians of all study participants, they were allowed to participate in the study. The study was conducted from March to September 2020.

### Study design and process

This study was conducted as a randomized controlled trial involving 50 adolescents. The selected subjects were randomly (block size = 3) assigned to experimental group 1 (G1, n = 16), experimental group 2 (G2, n = 17), and control group (CG, n = 17) through the R studio program (R studio desktop 1.2.5033). The randomization and subject registration procedures were conducted by the research director, and the subjects were conducted in a situation where they did not know which intervention they would receive. For all three groups, the PWI-SF, pulmonary function test (PFT), respiratory muscle pressure test (RMPT), and World Health Organization Quality of Life Scale Abbreviated Version (WHOQOL-BREF) were performed as a self-rating test of stress before and after the intervention. The intervention method was specific to each group. The intervention was given for 12 weeks, and data were collected from a total of 41 subjects, excluding 9 subjects who did not attend post-test. The flow of the study is shown in Fig 1.

### Intervention

**Experimental group 1.** The wind instrument and choral training group was guided to play a wind instrument and participate in a choral activity. The flute was the wind instrument used in this study as it was reported by a previous study to contribute to enhancing respiratory function [14]. The total time of the intervention was 1 h 30 min: 40 min of flute playing and 50-min of choral training. A different piece was selected each week for the flute and choral training to continuously induce the participants' interests. The intervention was given for 12 weeks in total, twice a week and 2 h per session. Participants were warmed up and cool down using breathing exercises for 10 min before and after the program started. The breathing training program teaches general abdominal breathing to suit vocal vocalization. A 10 min rest was

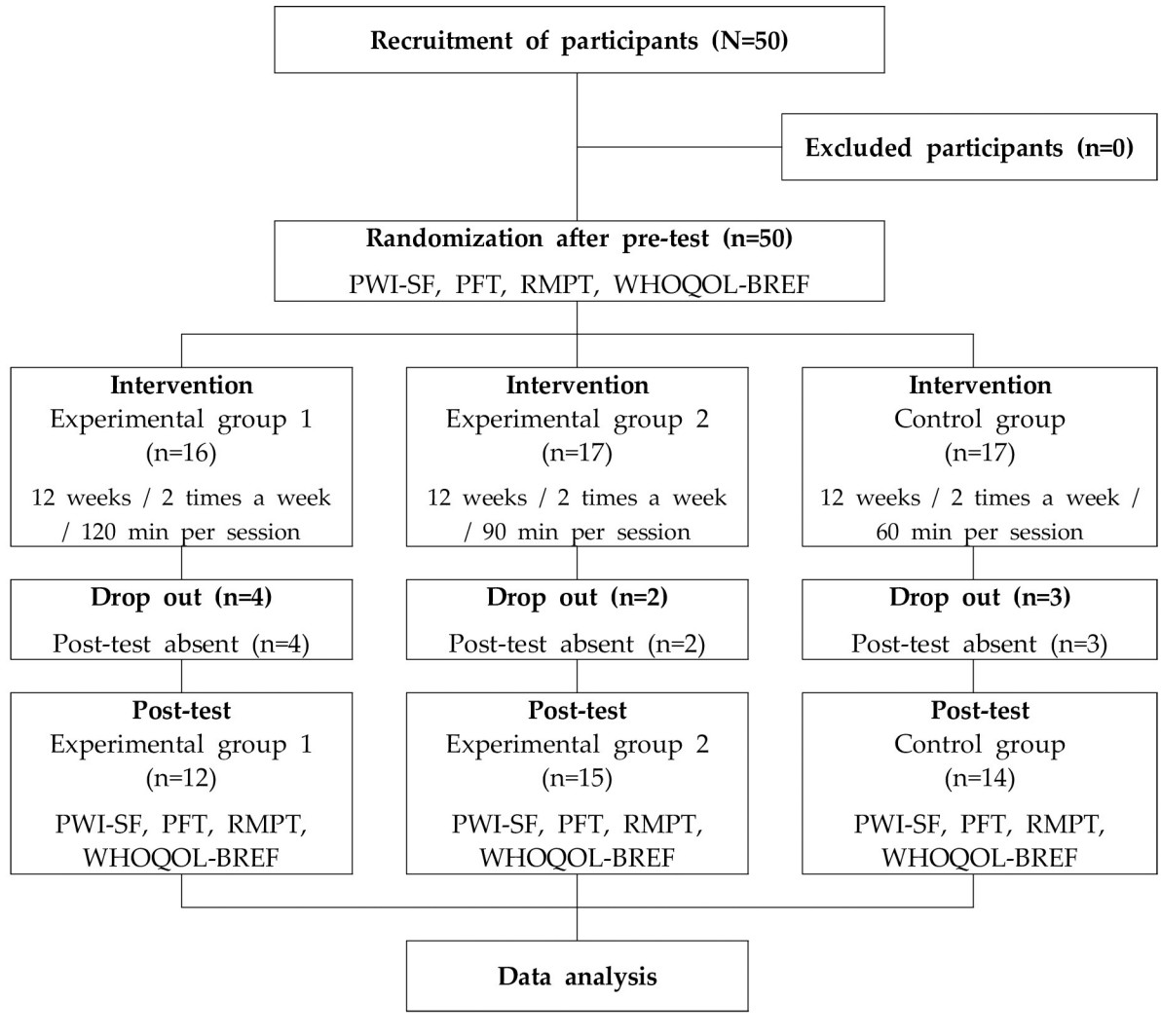

**Fig 1. Flow chart of this study.**

given after 40 min flute practice and after 40 min choral training. To maintain a high participation rate, a performance was held on the day of the last session, to which the participants' guardians and teachers were invited, in reference to a previous study (Fig 2) [11].

   **Experimental group 2.** The choral training group was guided to participate in a choral activity and receive a vocal training at the same time, in reference to a previous study [15]. The vocal training in this study involved basic abdominal breathing, thoracic breathing, and head voice methods as well as the training to produce high to low and short to long sounds. The program consisted of practice using songs. Participants were warmed up and cool down using breathing exercises for 10 minutes before and after the program started. The breathing training program teaches general abdominal breathing to suit vocal vocalization. The participants received vocal training for 30 min, followed by a 10 min rest, and after an additional 30 min vocal training, a choral program was performed. A different program was applied each week to continuously induce the participants' interests through 30 min training, 10 min rest, and another 30 min training. To maintain a high participation rate, a performance was held on the day of the last session for the participants' friends, guardians, and teachers (Fig 3).

| Time | 0~10 Min | 10~40 Min | 40~50 Min | 50~80 Min | 80~90 Min |
|---|---|---|---|---|---|
| 1 | Warm up Breathing training | Learn choreography 1 | Resting | Learn choreography 1 Retraining | Cool down Breathing training |
| 2 | Warm up Breathing training | Learn choreography 2 | Resting | Learn choreography 2 Retraining | Cool down Breathing training |
| 3 | Warm up Breathing training | Learn choreography 3 | Resting | Learn choreography 3 Retraining | Cool down Breathing training |
| 4 | Warm up Breathing training | Learn choreography 4 | Resting | Learn choreography 4 Retraining | Cool down Breathing training |
| 5 | Warm up Breathing training | Learn choreography 5 | Resting | Learn choreography 5 Retraining | Cool down Breathing training |
| 6 | Warm up Breathing training | Learn choreography 6 | Resting | Learn choreography 6 Retraining | Cool down Breathing training |
| 7 | Warm up Breathing training | Learn choreography 7 | Resting | Learn choreography 7 Retraining | Cool down Breathing training |
| 8 | Warm up Breathing training | Learn choreography 8 | Resting | Learn choreography 8 Retraining | Cool down Breathing training |
| 9 | Warm up Breathing training | Learn choreography 9 | Resting | Learn choreography 9 Retraining | Cool down Breathing training |
| 10 | Warm up Breathing training | Learn choreography 10 | Resting | Learn choreography 10 Retraining | Cool down Breathing training |
| 11 | Warm up Breathing training | Learn choreography 11 | Resting | Learn choreography 11 Retraining | Cool down Breathing training |
| 12 | Final performance | | | | |

**Fig 2. Wind instrument and choral music program.**

**Experimental group 3.** The control group was guided to participate in a general humanities program at the university, whereby they listened to music that helps with emotional relaxation. The program consisted of classical music and songs to promote emotional stability and increase the understanding of music through intermittent explanations. At the end of the experiment, the control group was allowed to participate in a program of their choice: the wind instrument and choral training program or the choral training alone. The intervention was given for 12 weeks in total, twice a week and 1 h per session.

## Outcome measures

**Psychological social well-being index short form.** The PWI-SF is a self-rating test of stress on a 4-point scale based on the general health questionnaires-60 (GHQ-60). The tool consists of 18 questions with a total score range of 0–54. Higher scores indicate higher levels of stress; scores $\geq 27$ indicate a high-risk stress group, scores of 9–26 indicate a potential stress group, and scores $\leq 8$ indicate a healthy group [13].

**Pulmonary function test.** The PFT is a health assessment for the respiratory system, whereby medical abnormality is tested based on the breathing flow volume and rate. The variables in this test are FVC, FEV1, and the ratio of the two as a percentage (FEV/FVC%), as well as peak expiratory flow (PEF) and maximal voluntary ventilation (MVV) [16].

**Respiratory muscle pressure test.** The RMPT allows an indirect measurement of the respiratory muscle strength based on pressure variation as muscle strength cannot be directly assessed. The maximal static pressure upon inhalation and exhalation is measured to assess the respiratory muscle strength. The measurements were taken in triplicate and the highest value

| Time | 0~10 Min | 10~50 Min | 50~60 Min | 60~110 Min | 110~120 Min |
|------|----------|-----------|-----------|------------|-------------|
| 1 | Warm up Breathing training | Wind instrument Program 1-1 (Do-re-mi song) | Resting | Learn choreography 1 | Cool down Breathing training |
| 2 | Warm up Breathing training | Wind instrument Program 2-1 (Jingle Bells) | Resting | Learn choreography 2 | Cool down Breathing training |
| 3 | Warm up Breathing training | Wind instrument Program 3-1 (Serenade to spring) | Resting | Learn choreography 3 | Cool down Breathing training |
| 4 | Warm up Breathing training | Wind instrument Program 4-1 (Cannon) | Resting | Learn choreography 4 | Cool down Breathing training |
| 5 | Warm up Breathing training | Wind instrument Program 5-1 (Faure Sicilienne) | Resting | Learn choreography 5 | Cool down Breathing training |
| 6 | Warm up Breathing training | Wind instrument Program 6-1 (Gariboldi Etudes OP 131 C major) | Resting | Learn choreography 6 | Cool down Breathing training |
| 7 | Warm up Breathing training | Wind instrument Program 7-1 (Gariboldi Etudes OP 131 A mimor) | Resting | Learn choreography 7 | Cool down Breathing training |
| 8 | Warm up Breathing training | Wind instrument Program 8-1 (Gariboldi Etudes OP 131 G major) | Resting | Learn choreography 8 | Cool down Breathing training |
| 9 | Warm up Breathing training | Wind instrument Program 9-1 (Gariboldi Etudes OP 131 C Minor) | Resting | Learn choreography 9 | Cool down Breathing training |
| 10 | Warm up Breathing training | Wind instrument Program 10-1 (Gariboldi Etudes OP 131 E Major) | Resting | Learn choreography 10 | Cool down Breathing training |
| 11 | Warm up Breathing training | Wind instrument Program 11-1 (Gariboldi Etudes OP 131 E Minor) | Resting | Learn choreography 11 | Cool down Breathing training |
| 12 | **Final performance** | | | | |

**Fig 3. Choral music program.**

was used in the analysis. The RMPT is expressed as the maximal expiratory pressure and maximal inspiratory pressure, with $cmH_2O$ as the unit of pressure [17].

**World Health Organization Quality of Life Scale Abbreviated Version.** The Korean version of the WHOQOL-BREF consists of 26 questions: seven questions on physical health, six questions on psychological health, three questions on social relationships, eight questions on lifestyle, and two questions on overall quality of life and general perception of health. Each question was rated on a Likert scale from 1 indicating "Strongly disagree" to 5 indicating "Strongly agree" and negative questions were reverse coded. The total score range was 26–130 with higher scores indicating higher quality of life [18].

**Data analysis.** Data analysis was performed using the SPSS version 25.0 (IBM, Chicago, IL). The general characteristics of subjects were presented with descriptive statistics based on the mean and standard deviation. The normality was tested using the Shapiro-Wilk test and a

normal distribution was found across all variables. To test the between-group homogeneity before the intervention, the χ2 test and one-way analysis of variance (ANOVA) were used. For the pretest-posttest comparison in each group, paired t-test was used. To assess the pattern of changes in the main effect and the group interaction according to time, a two-way ANOVA with repeated measures was used. The level of significance was set to α = 0.05.

## Results

### General characteristics of subjects in each group

Regarding the subjects' general characteristics, gender, age, height, weight, and BMI were homogeneous across groups (Table 1).

**Respiratory related factors.** The respiratory measurements before and after the intervention are presented in Table 2. All three groups showed no significant variation in the pre-intervention homogeneity test to confirm the homogeneity across groups. Among the main effects, the result for Time (F) showed significant variations in all measurements (p<0.01). However, except for maximal inspiratory pressure (MIP), all respiratory measurements exhibited an interaction to indicate a variation in the pattern of change.

**Mental and psychological factors.** The results of stress and quality of life measured before and after the intervention are presented in Table 3. All three groups showed no significant variation in the pre-intervention homogeneity test to confirm the homogeneity across groups. Among the main effects, the result for Time (F) showed significant variations in all measurements (p<0.01) and all measurements of stress and quality of life exhibited an interaction to indicate a variation in the pattern of change.

## Discussion

This study was conducted to determine the effects of a 12-week intervention combining wind instrument performance and a choral program on respiratory function, stress factors, and quality of life in adolescents residing in a metropolitan city with the intense negative effects of PM in the modern society.

Regarding respiratory function, with the exception of MEP, all measured variables exhibited an interaction to indicate a variation in the pattern of change. In the pretest-posttest comparison for G1 and G2, a significant variation in MEP was found; however, the lack of interaction is presumed to be because the overall intervention subjects did not have any pathological diseases in the respiratory tract and the interventions of G1 and G2 focused on the exhalation process. The correlation between lung capacity and muscle recruitment in flute performance showed that muscle activation during inhalation had less impact on muscle activation during exhalation, which is under a greater influence of dynamicity and lung capacity,

**Table 1. General characteristics of subjects in each group.**

|  | Experiment 1 group | Experiment 2 group | Control group | P-value |
|---|---|---|---|---|
| N | 12 | 15 | 14 | |
| Males | 9 (66%) | 10 (66%) | 7 (50%) | 0.39 |
| Age (years) | 18.42 (0.66) | 18.33 (0.61) | 18.21 (0.57) | 0.71 |
| Height (cm) | 169.75 (10.26) | 169.12 (6.15) | 167.03 (8.29) | 0.67 |
| Weight (kg) | 64.16 (12.08) | 64.86 (12.83) | 60.14 (10.30) | 0.53 |
| BMI (kg/m2 | 22.14 (2.85) | 22.67 (4.38) | 21.46 (2.45) | 0.63 |

Values are presented as mean (SD), BMI = Body Mass Index.

**Table 2. Comparison of respiratory related factors between the three groups.**

| Variables | | Experiment 1 (n = 12) | Experiment 2 (n = 15) | Control (n = 14) | P-value (Time × Group) |
|---|---|---|---|---|---|
| | | **Respiratory related factors** | | | |
| MIP | Pre | -82.87 (24.45) | -88.66 (32.81) | -84.24 (36.00) | 0.003 |
| | Post | -96.10 (29.74) | -100.85 (37.42) | -85.52 (32.91) | |
| | Post-Pre | 13.22 (8.47) | 12.18 (11.97) | 1.28 (5.97) | |
| | P-value | <0.001 | 0.001 | 0.43 | |
| MEP | Pre | 74.44 (20.21) | 76.20 (21.31) | 72.92 (41.94) | 0.288 |
| | Post | 88.56 (26.34) | 98.92 (34.49) | 85.80 (47.39) | |
| | Post-Pre | -14.12 (15.17) | -22.72 (21.63) | -12.87 (15.58) | |
| | P-value | 0.008 | 0.001 | 0.009 | |
| VC | Pre | 4.03 (0.80) | 3.91 (0.89) | 3.65 (0.77) | <0.001 |
| | Post | 4.55 (0.73) | 4.16 (0.84) | 3.67 (0.79) | |
| | Post-Pre | -0.52 (0.40) | -0.24 (0.21) | -0.01 (0.13) | |
| | P-value | <0.001 | <0.001 | 0.62 | |
| FVC | Pre | 4.01 (0.80) | 3.79 (0.88) | 3.63 (0.82) | 0.002 |
| | Post | 4.54 (0.73) | 4.15 (0.80) | 3.69 (0.76) | |
| | Post-Pre | -0.53 (0.39) | -0.35 (0.25) | -0.06 (0.28) | |
| | P-value | <0.001 | <0.001 | 0.42 | |
| FEV1 | Pre | 3.60 (0.83) | 3.50 (0.72) | 3.18 (0.73) | 0.003 |
| | Post | 4.08 (0.79) | 3.78 (0.79) | 3.19 (0.64) | |
| | Post-Pre | -0.48 (0.41) | -0.27 (0.18) | -0.00 (0.37) | |
| | P-value | 0.002 | <0.001 | 0.94 | |
| MVV | Pre | 117.65 (22.45) | 122.36 (28.98) | 106.10 (27.72) | 0.017 |
| | Post | 125.20 (20.91) | 124.80 (28.70) | 106.63 (27.95) | |
| | Post-Pre | -7.54 (6.07) | -2.44 (6.36) | -0.52 (5.71) | |
| | P-value | 0.001 | 0.15 | 0.73 | |

Values are presented as mean (SD), MIP = Maximal Inspiratory Pressure, MEP = Maximal Exspiratory Pressure, VC = Vital Capacity, FVC = Forced Vital Capacity, FEV1 = Forced Expiratory Volume in the First second, MVV = Maximal Voluntary Ventilation.

**Table 3. Comparison of pulmonary function levels between the three groups.**

| Variables | | Experiment 1 (n = 12) | Experiment 2 (n = 15) | Control (n = 14) | P-value (Time × Group) |
|---|---|---|---|---|---|
| | | **Mental and psychological factors** | | | |
| PWI-SF | Pre | 32.41 (6.44) | 33.40 (5.84) | 36.21 (6.38) | <0.001 |
| | Post | 15.50 (7.96) | 27.06 (6.91) | 33.50 (6.90) | |
| | Post-Pre | 16.91 (4.42) | 6.33 (6.65) | 2.71 (3.56) | |
| | P-value | <0.001 | 0.002 | 0.014 | |
| WHO QoL | Pre | 81.58 (14.49) | 82.53 (14.43) | 87.28 (17.50) | 0.001 |
| | Post | 99.75 (15.85) | 89.46 (12.96) | 93.71 (17.92) | |
| | Post-Pre | -18.16 (12.17) | -6.93 (6.35) | -6.42 (5.40) | |
| | P-value | <0.001 | <0.001 | 0.008 | |

Values are presented as mean (SD), PWI-SF = Psychological social Well-being Index Short-Form, WHOQoL = World Health Organization Quality of Life scale abbreviated version.

while substantial individual variations in the muscle recruitment during inhalation have been reported [19]. However, in this study, it is considered that choral training and basic diaphragm breathing as well as wind instrument performance improved overall respiratory capacity and inspiratory capacity. Compared to a breathing training using an incentive spirometer, wind instrument performance was shown to be more effective in a previous study [20]. In a study applying block flute for 1 month in pediatric and adolescent patients with asthma, a significant improvement was found in the experimental group, and surprisingly, the control group of individuals with no acute or chronic disease also showed a significant improvement [21]. Wind instruments such as flute, saxophone, and the bassoon require a technique to blow accurately through the mouthpiece, which necessitates more precise working in the process of breathing so as to contribute in enhancing respiratory function [22]. Wind instruments produce the sound as vibration is applied to the air blowing through the pipe, and they ensure an effective way of breathing training through abdominal breathing with rapid and deep inhalation and long exhalation via pursed lips as natural movements and assist with diaphragmatic breathing to increase the airway pressure during exhalation [11, 23]. At this time, it is thought that the inhalation process of trying to breathe in quickly had an effect on the strengthening of the respiratory muscles.

The measurements regarding stress factors and quality of life in this study showed that all measured variables exhibited an interaction to indicate a variation in the pattern of change. Also, in the post-hoc analysis, G1 showed a significant improvement compared to the other groups. Music-based rehabilitation programs have been reported to exert positive effects on the motivation, participation, and the emotional state of participants in comparison to non-music-based programs. Such changes in emotional aspects have also been shown to relay to potential rehabilitation effects [24]. Music-based programs are effective in reducing the disturbance and anxiety of breathing in patients with a respiratory disease, which has been reported to improve the quality of sleep as well as physiological parameters [25]. In this study, the tasks in the music-based program were adjusted according to the individual functional levels of the participants; the immediate feedback in line with the attempts of self-breathing exercise was presumed to have reduced the burden of program participation and the possible negative emotions (such as depressive feelings) [26]. This study is the first study to investigate the effect of a music program combining wind instrument and choral training on respiratory function and stress factors.

This study had limitations. First, although the sample size was estimated using the G power program, the number of participants was too small to present clinical data. Second, the deviation in the intervention time between the two experimental groups implies room for improvement in the intervention criteria. Third, the individual differences in the competence of playing the instrument and vocal abilities in choral activities could have prevented the intervention process from generalization. Fourth, as the participants were adolescents, various activities including leisure and hobbies could not be completely controlled. Lastly, as the participants were healthy adolescents who had few or no respiratory problems, it may be difficult to generalize the results to patients with respiratory symptoms in clinical practice. Despite the limitations, the findings in this study form the basis for further investigations regarding increased PM-related air pollution in the modern and future society.

## Conclusions

This study investigated the effects of an intervention combining wind instrument performance and a choral program on respiratory functions, stress factors, and quality of life in adolescents residing in a metropolitan city with the intense negative effects of PM in the modern society.

The results showed that the 12-week intervention had positive effects in improving respiratory function, stress factors, and quality of life in adolescents. Therefore, the study findings are expected to support future studies on the promotion of overall health including respiratory function and psychological factors through various music-based programs. Furthermore, future studies should be conducted on patients with reduced respiratory function in clinical settings.

## Supporting information

**S1 Checklist. CONSORT 2010 checklist of information to include when reporting a randomised trial\*.**
(DOC)

**S1 File.**
(PDF)

**S2 File.**
(PDF)

## Author Contributions

**Conceptualization:** Byeong Soo Kim, Ho Kim, Ji Youn Kim.

**Data curation:** Byeong Soo Kim, Ho Kim, Ji Youn Kim.

**Formal analysis:** Byeong Soo Kim, Ho Kim, Ji Youn Kim.

**Funding acquisition:** Ji Youn Kim.

**Investigation:** Byeong Soo Kim, Ho Kim, Ji Youn Kim.

**Methodology:** Byeong Soo Kim, Ho Kim, Ji Youn Kim.

**Project administration:** Byeong Soo Kim, Ho Kim, Ji Youn Kim.

**Resources:** Ho Kim, Ji Youn Kim.

**Visualization:** Ji Youn Kim.

**Writing – original draft:** Byeong Soo Kim, Ho Kim, Ji Youn Kim.

**Writing – review & editing:** Byeong Soo Kim, Ho Kim, Ji Youn Kim.

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
