## [Decision Letter · Decision Letter 0]

31 Oct 2022

PONE-D-22-27071Effects of a choral program combining wind instrument performance and breathing training on respiratory function, stress, and quality of life in adolescents: A randomized controlled trialPLOS ONE

Dear Dr. Kim,

Thank you for submitting your manuscript to PLOS ONE. After careful consideration, we feel that it has merit but does not fully meet PLOS ONE’s publication criteria as it currently stands. Therefore, we invite you to submit a revised version of the manuscript that addresses the points raised during the review process.

We look forward to receiving your revised manuscript.

Kind regards,

Walid Kamal Abdelbasset, Ph.D.

Academic Editor

PLOS ONE

Journal Requirements:

"Yes.This research was supported by the National Research Foundation of Korea(NRF) grant funded by the Korea government(MSIT)(No.2019R1G1A110054313)"

Please state what role the funders took in the study.  If the funders had no role, please state: ""The funders had no role in study design, data collection and analysis, decision to publish, or preparation of the manuscript."" If this statement is not correct you must amend it as needed. 

"No"

5. Please ensure that you refer to Figures 2 and 3 in your text as, if accepted, production will need this reference to link the reader to the figure.

6. Please upload a new copy of Figures 1 to 3 as the detail is not clear. Please follow the link for more information:

https://blogs.plos.org/plos/2019/06/looking-good-tips-for-creating-your-plos-figures-graphics/

https://blogs.plos.org/plos/2019/06/looking-good-tips-for-creating-your-plos-figures-graphics/

Reviewers' comments:

Reviewer's Responses to Questions

**Comments to the Author**

1. Is the manuscript technically sound, and do the data support the conclusions?

Reviewer #1: No

Reviewer #2: Partly

Reviewer #3: Partly

2. Has the statistical analysis been performed appropriately and rigorously? 

Reviewer #1: I Don't Know

Reviewer #2: No

Reviewer #3: I Don't Know

3. Have the authors made all data underlying the findings in their manuscript fully available?

Reviewer #1: No

Reviewer #2: Yes

Reviewer #3: Yes

4. Is the manuscript presented in an intelligible fashion and written in standard English?

Reviewer #1: Yes

Reviewer #2: Yes

Reviewer #3: No

5. Review Comments to the Author

Reviewer #1: Thanks a lot for your effort but ,let me understand the following questions and comments

What is the novelty of your research?

What’s the clinical purpose of the research?

Where is the hypothesis

This manuscript requires a significant amount of improvement in:

Abstract:

- you should write subtitle for each part of Abstract

- The background needs to be shortened.

- Methods section is poorly framed. It has to be re-written.

- Demographic profile of patients is not mentioned.

Method

-what is the Type of randomization; details of any restriction (such as blocking and block size)

-what is the Mechanism used to implement the random allocation sequence (such as sequentially numbered containers), describing any steps taken to conceal the sequence until interventions were assigned

-Who generated the random allocation sequence, who enrolled participants, and who assigned participants to interventions

-Blinding If done, who was blinded after assignment to interventions (for example, participants, care providers, those assessing outcomes) and how

-Participants were warmed up and cool down using breathing exercises for 10 min before and after the program started (which types of breathing and how the participant did that and why you select that types )

Discussion:

The discussion section needs to be described scientifically. Kindly frame it along the following lines:

1-Main findings of the present study

2-Comparison with other studies

2-Implication and explanation of findings

-Describe sources of potential bias and imprecision.

-Generalisability of the trial findings need to be put.

The section needs to be as per well defined objectives.

- It has to be framed in such a way that readers are able to have good understanding of the current evidences and rationale of the paper.

Figures not clear for reading example

Fig 3. Choral music program

Fig 2. Wind instrument and choral music program

thanks regards

Reviewer #2: The term << breathing training>> in the title doesn't explore the act intervention in the study , as it was done through the two method of intervention not in separated intervention. The author doesn't mention body mass index as one of the inclusion criteria for the selected participants. Small sample size as sample size calculation wasn't done.

Reviewer #3: Abstract:

It is better to formulate abstract in a structured manner

Methods section is poorly framed. It has to be re-written.

Demographic profile of patients is not mentioned.

Briefly mention your study findings

Add key words

Introduction:

Explain the rationale of the study. Kindly focus on three elements of introduction.

a. What is known about the topic? (Background)

b. What is not known? (The research problem)

c. Why the study was done? (Justification)

Methods

How was sample size determined?

It is better to name grous by intervention techniques,not experimental group

What was the method used to generate the random allocation sequence?

Explain the type of randomization.

Was there any restriction like blocking and block size?

What kind of mechanism was used to implement the random allocation sequence?

Were any steps taken to conceal the sequence until interventions were assigned?

Who generated the random allocation sequence?

Who enrolled participants?

Who assigned participants to interventions?

What methods were used for analysis of drop out cases

Results

a. Results need to provide answers to the questions raised/researchable problem

b. Results need to follow ABC (accuracy, brevity, clarity)

The discussion section needs to be described scientifically. Kindly frame it along the following lines:

i. Main findings of the present study

ii. Comparison with other studies

iii. Implication and explanation of findings

iv. Strengths and limitations

v. Conclusion, recommendation and future direction.

2. Explain in short about trial limitations.

3. Describe sources of potential bias and imprecision.

6. PLOS authors have the option to publish the peer review history of their article (what does this mean?). If published, this will include your full peer review and any attached files.

Reviewer #1: No

Reviewer #2: No

Reviewer #3: No

---

## [Author Response · Author response to Decision Letter 0]

10 Jan 2023

Dear Reviewers

Our research team faithfully responded to the request, and we are willing to actively respond to additional reviews.

Thank you for reviewing our article, and we will contribute to academic promotion through better research in the future.

thank you

-Best regards

---

## [Decision Letter · Decision Letter 1]

24 Mar 2023

PONE-D-22-27071R1Effects of a choral program combining wind instrument performance and breathing training on respiratory function, stress, and quality of life in adolescents: A randomized controlled trialPLOS ONE

Dear Dr. Kim,

Thank you for submitting your manuscript to PLOS ONE. After careful consideration, we feel that it has merit but does not fully meet PLOS ONE’s publication criteria as it currently stands. Therefore, we invite you to submit a revised version of the manuscript that addresses the points raised during the review process.

Your manuscript has been assessed by two of the previous reviewers, and additionally by a statistical reviewer; their comments are available below. While the previous reviewers are satisfied with the revisions made, the statistical reviewer (Reviewer #4) has raised concerns which we ask you to address in further revisions, particularly regarding interpretation of significant interaction effects. Please ensure you address each of the reviewer's comments when revising your manuscript.

We look forward to receiving your revised manuscript.

Kind regards,

Hugh Cowley

Staff Editor

PLOS ONE

Reviewers' comments:

Reviewer's Responses to Questions

**Comments to the Author**

1. If the authors have adequately addressed your comments raised in a previous round of review and you feel that this manuscript is now acceptable for publication, you may indicate that here to bypass the “Comments to the Author” section, enter your conflict of interest statement in the “Confidential to Editor” section, and submit your "Accept" recommendation.

Reviewer #1: All comments have been addressed

Reviewer #2: All comments have been addressed

Reviewer #4: (No Response)

2. Is the manuscript technically sound, and do the data support the conclusions?

Reviewer #1: Yes

Reviewer #2: Yes

Reviewer #4: Yes

3. Has the statistical analysis been performed appropriately and rigorously? 

Reviewer #1: Yes

Reviewer #2: Yes

Reviewer #4: No

4. Have the authors made all data underlying the findings in their manuscript fully available?

Reviewer #1: Yes

Reviewer #2: Yes

Reviewer #4: No

5. Is the manuscript presented in an intelligible fashion and written in standard English?

Reviewer #1: Yes

Reviewer #2: Yes

Reviewer #4: Yes

6. Review Comments to the Author

Reviewer #1: thanks alot for your response

Reviewer #2: Thanks for your efforts in editing the previous comments which had been done appropriately. The results regarding the manuscript will add to knowledge.

Reviewer #4: A three-arm randomized controlled trial was conducted which aimed to assess the effects of a 12-week intervention on stress, quality of life, and respiratory function in adolescents with exposure to air pollution. Arm by time interactions were significant in relation to respiratory function, with the exception of maximal inspiratory pressure. Additionally, significant interactions of arm by time were observed for stress and quality of life factors.

Major revisions:

1- Tables 2 and 3: If the interaction effects are significant, provide an interpretation of these results, but do not test main effects because the tests for main effects are uninteresting in light of significant interactions. If interaction effects are non-significant, drop the interaction effects from the model and test the main effects. Determining which results to present when testing interactions is often a multi-step process.

2- Tables 2 and 3: Provide p-values instead of t- and F-values.

3- Provide meaningful interpretations of significant interaction effects.

Minor revisions:

1- Indicate the date range subjects were enrolled in the study.

2- Line 101: If block randomization was used, indicate the block size.

3- State and justify the study’s target sample size with a pre-study statistical power calculation. The power calculation should include: (1) the estimated outcomes in each group; (2) the α (type I) error level; (3) the statistical power (or the β (type II) error level); (4) the target sample size and (5) for continuous outcomes, the standard deviation of the measurements.

4- Table 1:

- In addition to the frequency, provide the percent female.

- In the last column, provide p-values instead of F/chi-square.

7. PLOS authors have the option to publish the peer review history of their article (what does this mean?). If published, this will include your full peer review and any attached files.

Reviewer #1: No

Reviewer #2: No

Reviewer #4: No

---

## [Author Response · Author response to Decision Letter 1]

6 Apr 2023

Before responding to the review comments, thank you for your interest in our study.

Reviewer 1

Thanks a lot for your help in the revision of my thesis. May God bless you always...

Reviewer 2

Thanks a lot for your help in the revision of my thesis. May God bless you always...

Reviewer 4

<Major revisions>

1) Tables 2 and 3: If the interaction effects are significant, provide an interpretation of these results, but do not test main effects because the tests for main effects are uninteresting in light of significant interactions. If interaction effects are non-significant, drop the interaction effects from the model and test the main effects. Determining which results to present when testing interactions is often a multi-step process.

A: We have accepted your request and completed the correction.

2) Tables 2 and 3: Provide p-values instead of t- and F-values.

A: We have accepted your request and completed the correction.

3) Provide meaningful interpretations of significant interaction effects.

A: Dear reviewer, I think that the meaning of interaction has been sufficiently provided in the discussion. If there are any deficiencies, we will proceed with additional corrections later. Thanks a lot.

<Minor revisions>

1) Indicate the date range subjects were enrolled in the study.

A: We have accepted your request and completed the correction.

2) Line 101: If block randomization was used, indicate the block size.

A: We have accepted your request and completed the correction.

3) State and justify the study’s target sample size with a pre-study statistical power calculation. The power calculation should include: (1) the estimated outcomes in each group; (2) the α (type I) error level; (3) the statistical power (or the β (type II) error level); (4) the target sample size and (5) for continuous outcomes, the standard deviation of the measurements.

A: We have accepted your request and completed the correction.

4) Table 1:

- In addition to the frequency, provide the percent female.

- In the last column, provide p-values instead of F/chi-square.

A: We have accepted your request and completed the correction.

With the above, we will send the responses to the reviewers' review opinions of this researcher.

I would like to thank this researcher for helping me to publish a better study, and I would appreciate it if you would always give me advice if there are any deficiencies in corrections.

P.s. God bless you always

---

## [Decision Letter · Decision Letter 2]

11 Sep 2023

PONE-D-22-27071R2Effects of a choral program combining wind instrument performance and breathing training on respiratory function, stress, and quality of life in adolescents: A randomized controlled trialPLOS ONE

Dear Dr. Kim,

Thank you for submitting your manuscript to PLOS ONE. After careful consideration, we feel that it has merit but does not fully meet PLOS ONE’s publication criteria as it currently stands. Therefore, we invite you to submit a revised version of the manuscript that addresses the points raised during the review process.

Three reviewers have reassessed the manuscript. Whilst they are overall satisfied with the revisions, reviewer 4 has provided some very minor comments to address.

We look forward to receiving your revised manuscript.

Kind regards,

Hanna Landenmark

Staff Editor

PLOS ONE

Journal Requirements:

Reviewers' comments:

Reviewer's Responses to Questions

**Comments to the Author**

1. If the authors have adequately addressed your comments raised in a previous round of review and you feel that this manuscript is now acceptable for publication, you may indicate that here to bypass the “Comments to the Author” section, enter your conflict of interest statement in the “Confidential to Editor” section, and submit your "Accept" recommendation.

Reviewer #1: All comments have been addressed

Reviewer #2: All comments have been addressed

Reviewer #4: (No Response)

2. Is the manuscript technically sound, and do the data support the conclusions?

Reviewer #1: Yes

Reviewer #2: Yes

Reviewer #4: Yes

3. Has the statistical analysis been performed appropriately and rigorously? 

Reviewer #1: Yes

Reviewer #2: Yes

Reviewer #4: Yes

4. Have the authors made all data underlying the findings in their manuscript fully available?

Reviewer #1: Yes

Reviewer #2: Yes

Reviewer #4: No

5. Is the manuscript presented in an intelligible fashion and written in standard English?

Reviewer #1: Yes

Reviewer #2: Yes

Reviewer #4: Yes

6. Review Comments to the Author

Reviewer #1: thanks for your replay

Reviewer #2: The author has addressed all the required comments and I recommend the manuscript gone to further process for publication

Reviewer #4: Minor revisions:

1- Line 97: This sentence is poorly worded. "Through the screening process, 0 subjects were excluded as they dissatisfied the inclusion criteria or satisfied the exclusion criteria, and a total of 50 subjects participated in this study." Consider the following instead. "During the screening process, all subjects satisfied the inclusion and exclusion criteria. A total of 50 subjects participated in the study."

2- Simplify line 101: The study was conducted from March to September 2020.

Note: Line numbers refer to those in the tracked changes version of revision 2.

7. PLOS authors have the option to publish the peer review history of their article (what does this mean?). If published, this will include your full peer review and any attached files.

Reviewer #1: No

Reviewer #2: No

Reviewer #4: No

---

## [Author Response · Author response to Decision Letter 2]

6 Oct 2023

Before responding to the review comments, thank you for your interest in our study.

Reviewer 1

Thanks a lot for your help in the revision of my thesis. May God bless you always...

Reviewer 2

Thanks a lot for your help in the revision of my thesis. May God bless you always...

Reviewer 4

<Minor revisions>

1) 1- Line 97: This sentence is poorly worded. "Through the screening process, 0 subjects were excluded as they dissatisfied the inclusion criteria or satisfied the exclusion criteria, and a total of 50 subjects participated in this study." Consider the following instead. "During the screening process, all subjects satisfied the inclusion and exclusion criteria. A total of 50 subjects participated in the study."

A: We have accepted your request and completed the correction

2) 2- Simplify line 101: The study was conducted from March to September 2020.

A: We have accepted your request and completed the correction

With the above, we will send the responses to the reviewers' review opinions of this researcher.

I would like to thank this researcher for helping me to publish a better study, and I would appreciate it if you would always give me advice if there are any deficiencies in corrections.

P.s. God bless you always

---

## [Decision Letter · Decision Letter 3]

25 Oct 2023

Effects of a choral program combining wind instrument performance and breathing training on respiratory function, stress, and quality of life in adolescents: A randomized controlled trial

PONE-D-22-27071R3

Dear Dr. Kim,

We’re pleased to inform you that your manuscript has been judged scientifically suitable for publication and will be formally accepted for publication once it meets all outstanding technical requirements.

Kind regards,

Andrea Martinuzzi

Academic Editor

PLOS ONE

Additional Editor Comments (optional):

Reviewers' comments:

Reviewer's Responses to Questions

**Comments to the Author**

1. If the authors have adequately addressed your comments raised in a previous round of review and you feel that this manuscript is now acceptable for publication, you may indicate that here to bypass the “Comments to the Author” section, enter your conflict of interest statement in the “Confidential to Editor” section, and submit your "Accept" recommendation.

Reviewer #1: All comments have been addressed

Reviewer #2: All comments have been addressed

Reviewer #4: All comments have been addressed

2. Is the manuscript technically sound, and do the data support the conclusions?

Reviewer #1: Yes

Reviewer #2: Yes

Reviewer #4: (No Response)

3. Has the statistical analysis been performed appropriately and rigorously? 

Reviewer #1: Yes

Reviewer #2: Yes

Reviewer #4: (No Response)

4. Have the authors made all data underlying the findings in their manuscript fully available?

Reviewer #1: Yes

Reviewer #2: Yes

Reviewer #4: (No Response)

5. Is the manuscript presented in an intelligible fashion and written in standard English?

Reviewer #1: Yes

Reviewer #2: Yes

Reviewer #4: (No Response)

6. Review Comments to the Author

Reviewer #1: thanks alot for your replay

Reviewer #2: Thanks for addressing the previously mentioned comments. The manuscript seems to consumed a hard efforts

Reviewer #4: (No Response)

7. PLOS authors have the option to publish the peer review history of their article (what does this mean?). If published, this will include your full peer review and any attached files.

Reviewer #1: No

Reviewer #2: **Yes: **Tamer I. Abo Elyazed, Ass.prof of Physical Therapy, Beni-Suef University

Reviewer #4: No

---

## [Editor Report · Acceptance letter]

30 Oct 2023

PONE-D-22-27071R3 

Effects of a choral program combining wind instrument performance and breathing training on respiratory function, stress, and quality of life in adolescents: A randomized controlled trial 

Dear Dr. Kim:

I'm pleased to inform you that your manuscript has been deemed suitable for publication in PLOS ONE. Congratulations! Your manuscript is now with our production department. 

Kind regards, 

on behalf of

Dr. Andrea Martinuzzi 

Academic Editor

PLOS ONE